# New Antifungal Microbial Pigment Applied to Improve Safety and Quality of Processed Meat-Products

**DOI:** 10.3390/microorganisms9050989

**Published:** 2021-05-04

**Authors:** Hatem Ali Salama, Ahmed Noah Badr, Manal F. Elkhadragy, Ahmed Mohamed Said Hussein, Ibrahim Abdel-Salam Shaban, Hany M. Yehia

**Affiliations:** 1Food Science and Nutrition Department, College of Food Science and Agriculture, King Saud University, Riyadh 11451, Saudi Arabia; haali@ksu.edu.sa (H.A.S.); hanyehia@ksu.edu.sa (H.M.Y.); 2Department of Food Technology, National Research Center, Dokki, Cairo 12622, Egypt; a.said22220@yahoo.com; 3Department of Food Toxicology and Contaminants, National Research Center, Dokki, Cairo 12622, Egypt; 4Department of Biology, Faculty of Science, Princess Nourah Bint Abdulrahman University, Riyadh 11564, Saudi Arabia; mfelkhadragy@pnu.edu.sa; 5Department of Chemistry of Natural and Microbial Product, National Research Center, Dokki, Cairo 12622, Egypt; abdelsalam66@hotmail.com; 6Department of Food Science and Nutrition, Faculty of Home Economics, Helwan University, Cairo 11221, Egypt

**Keywords:** antifungal pigment, minced meat, organoleptic properties, bioactive components, color improvement, shelf life extension, *Penicillium purpurogenum*

## Abstract

Minced meat is involved within numerous products, where their color attributes are affected by consumer preferences. This study was aimed to ameliorate processed meat color, using a microbial red pigment. Antibacterial, antifungal, citrinin-free, and toxicity of pigment were determined. Meatballs and burgers were manufactured using pigment at 3 mg/g of meat. Texture, color, shelf life extension, and organoleptic properties were estimated for treated meats. Results were expressed by a real antimicrobial for pigment, even via several extracting systems. The MIC and MFC of pigment were 320 µg/g and 2.75 mg/g media, respectively. Bioactive components of pigment were detected using the GC–MS and the FTIR apparatus. The bioactive carbohydrates include oligo and polysaccharides were manifested with real curves. Secretion of ochratoxin A and aflatoxins in fungal media receives pigment was decreased by up to 54% and 45%, respectively. The presence of bioactive carbohydrates may trap mycotoxin out of the recovered amounts. The manufactured products were enhanced for their color and taste with fine texture changes. The shelf life of colored-frying meat was recorded by an extension compared to the control. In conclusion, the results were recommended microbial red-pigment implementation in meats manufacturing for ameliorating recorded of color, as antimycotoxigenic, and shelf life extension.

## 1. Introduction

During the last decades, a great interest has occurred for alternative pigments, which are obtained from natural sources. This resulted mainly because of synthetic dye issues. Disadvantages of synthetic colorants were reported destroying food nutrients, cause hazards to consumers, and led to health problems. The most notable health-hazards were included cancer occurrence, sudden mood swings, hyperactivity in children, DNA damage or genotoxicity, attention-deficit disorder, and detrimental effects on the environment [1]. Natural colorants such as anthocyanins originated from plants that are not available throughout the year. Additionally, it becomes faint and unstable against the effect of heat, light, and pH of the medium [2].

Minced meat is a meat type that widely inserts in burgers, hawawchi, meatballs, luncheons, and sausage. Pigments usually apply to improve products color. More dangerously the chemical additives may be used for the same target, like nitrous salts utilization [3]. A great demand for natural and reliable colorant sources was recently increased. The color instability and production-leakage of pigments sourced from plants or animals represent real challenges that limit the application [4]. The microbial colorant production is advantageous due to the rapid growth of the microorganism, which leads to a high production rate of non-toxic ecofriendly compounds. Several authors have reported series of microorganisms as a producer of large quantities of colorant involving, *Monascus* sp., *Cordyceps, Serratia, Streptomyces, Penicillium atrovenetum*, and *Penicillium herquei* [5,6].

The microbial colorants could participate in a shelf life extension of food. This will achieve an additional function in meat products and be deemed as an alternative to chemical preservatives hazards. Several types of natural components, including extracts, was reported by in-vivo antimycotoxigenic impacts, and has a capacity to reduce their risk [7]. The antimicrobial function was reported for the pigments that were sourced from microorganisms in numerous research [8]. Minced meat could contaminate by mycotoxins regarded various factors like transferred to the tissues from contaminant feed, the spices utilized in product-recipes; particularly aflatoxins (AFs), cyclopiazonic acid (CYA), and ochratoxin A (OCA) [9]. In this regard, the enrichment of the product recipe by bioactive molecules (phenolic, oligo and polysaccharides, dietary fibers) may play a role as a trap of mycotoxins [10].

The study was aimed to enhance the meat-product color by a microbial red pigment that could achieve safety characteristics. Color is an important factor joined to the consumers’ preferences, meat products will choice more increased if their color was vivid and brightened. In this way, the pigment produced by the *Penicillium* strain was applied in the manufacturing of minced-meat products. It will be beneficial if the pigment application could achieve several targets for the final products, including shelf-life extension and safety. For this reason, the antibacterial, antifungal, and antimycotoxigenic impacts of pigment were evaluated as additional benefits.

## 2. Materials and Methods

### 2.1. Microorganism

The culture of *Penicillium purpurogenum* (2603), in 250 mL Erlenmeyer flasks, was inoculated and was utilized to produce a red pigment according to the media and optimum condition described by Ahmed et al. [11]. Briefly, the spore suspension (2 × 10^7^ spores/mL) was used to inoculate to the Czapeck yeast agar (CYA) growth media, which contained the following ingredients (g/L): glucose (30), yeast extract (2), peptone (10), NaNO_3_ (3), KCl (0.5), MgSO_4_ (0.5), and agar (25), incubated (7 days/28 °C) then stored (4 °C).

The bacterial strains of the antimicrobial assay included *Bacillus cereus* ATCC 4342; *Escherichia coli* ATCC 11229; *Staphylococcus aureus* NCTC 10788; and *Salmonella typhi* ATCC 14028 purchased from Egyptian Microbial Culture Collection (EMCC), Ain Shams, Egypt. Four toxigenic fungi were used for antifungal assays, including *Aspergillus flavus* ITEM 698, *A. ochraceus* ITEM 5010, *A. niger* ITEM 3856, and *Fusarium culmorum* KF846, that were obtained from Agrofood Microbial Culture Collection (ITEM), ISPA, Bari, Italy.

### 2.2. Pigments Extraction

The culture was shaken at 200 rpm (28 °C/7 days). The production medium was prepared using the following nutrients (g/L distilled water): Mannitol (20), NaNO_3_ (3.0), KCl (0.5), MgSO_4_.7H_2_O (0.5), KH_2_PO4 (1.0), and FeSO_4_.7H_2_O (0.1). Cultures from Dox production media were filtered through filter paper (No. 1; Whatman), the biomass was dried at 50 °C for 48 h and weighed. The red pigment was extracted with chloroform, methanol, isopropanol, and ethyl acetate. The extract was concentrated till dryness and dissolved in 10 mL distilled water for application in characteristics evaluations [11]. The red pigment was further extracted by isopropanol for food applications experiments, evaporated, and concentrated until dryness using the rotary evaporator (Heidolph, Hei-VAP Platinum 5; ProfiLab24 GmbH; Landsberger; Germany. The absorbance was read at 630 using a UV-visible spectrophotometer (Spectro UV-2505; Labomed Co., Int., La Cienega Blvd, Los Angeles, CA, USA).

### 2.3. Determination of Pigment Antimicrobial Properties

The antimicrobial properties of the pigment were determined using several assays to ensure its efficiency. Firstly, the minimal inhibitory concentration (MIC) and the minimum antifungal concentration (MFC) were evaluated as the lowest concentration affected the bacteria and fungi growth rates, respectively. The MIC was determined as the lowest concentration at which no growth occurs, it was done as a method described by Sakanaka et al. [12]. Azithromycin was applied as a positive control. Regarding the MFC, fungal-growth inhibition was tested to determine the impact of pigment as previously described by Abdel Razek et al., 2017 [13]. The concentration required to give 50% inhibition of hyphen growth IC_50_ was calculated from the regression equation, where mycostatin was used as a positive control. The antibacterial antifungal characteristics of red pigment also were determined using the diffusion assay as the methodology described previously [14,15].

### 2.4. Determination of Pigment Impact on Mycotoxin

In broth media of fungal growth, antimycotoxin properties of the red pigment were evaluated according to the method described by Shehata et al. [16], using the pigment extract (1, 3, and 5 mg/mL) against fungal growth of *Aspergillus flavus* ITEM 698 (AFs producer)*,* and *A. ochraceus* ITEM 5010 (OCA producer). Briefly; in conical flasks contains autoclaved 150 mL of yeast extract sucrose media (YES) for fungal growth of applied strain. Each conical flask was inoculated by the fungal spores that was resuspended in tween/water solution at a concentration of 10^5^ spores/mL. Nine flasks were prepared for the three applied concentration of pigment extracts (1, 3, and 5 mg/mL; three replicates per each) in comparison to the control flask, which were only inoculated by fungal spores. This design was applied for the two fungal types under the investigation. The reduction of mycelial growth weight will reflect the antifungal activates, while the toxin producing reduction reflect the safety impact of the applied concentration. The AFs and OCA were recovered from the growth media, cleaned up using Aflatest and Ochratest columns, respectively. The reduction of mycotoxins joined with the pigment was determined according to the conditions, columns, and apparatus described in Shehata et al. [16].

### 2.5. Preparation of Pigment for Toxicity Determination

The dried pigment was extracted successively using four different solvents to evaluate the cytotoxicity of polar and non-polar components. Solvent that were applied were petroleum ether, chloroform, isopropanol, and water. The pigment was soaked in a shaking water bath overnight under aseptic conditions then filtered using Whatman filter paper No.1. The extracts were concentrated under reduced pressure and temperature (40 °C) using a rotary evaporator.

### 2.6. Brine Shrimp Lethality Bioassay

Brine shrimp lethality bioassay (BSLB) was carried out to investigate pigment cytotoxicity. The BLSB gives evidence predictive of cytotoxicity and harmful activity. It was applied successfully as a registered study of cytotoxic and antitumor agents [17]. Brine shrimp eggs were collected from Ramsis Damian AquaLab, Hurgada, Egypt, as a gift sample for the research work. It was hatched in a vessel filled with simulated sterile artificial seawater (38 g/L NaOH/ pH 8.5). A yeast solution (0.06%) was added to the hatching vessel for feeding requirements [18], then were incubated in the dark (28 °C/36 h/constant aeration), active larvae were transferred (each ten) by capillary pastier pipette to a tube containing 5 mL of seawater in the light condition [19]. Where each tube was used for one concentration and one extract. The surviving larvae (deaths) were counted after incubating (25 °C/24 h) and lethality concentration LC_50_ was assessed.

### 2.7. Determination of Citrinin Residues in Pigment

The HPTLC technique was applied to detect the presence of citrinin in red pigment as described by Ahmed et al., 2010 [20]. Briefly, by utilizing a CAMAG (Muttenz, Switzerland) Linomat V sample applicator with the 100-µL syringe. The sample was performed on 10 cm × 10 cm HPTLC plates and inserted into a chamber that was previously saturated by the mobile phase. The mobile phase was applied in linear ascending development of toluene: ethyl acetate: formic acid as 3:2:1 (*v/v*), respectively. The running system condition was adjusted (27 °C/65% RH) through the running time, and the scanning samples were done using the same condition reported before [20]. An injection of 1 µL citrinin was performed as a standard for the calibration test.

### 2.8. The Evaluation of Red Pigment by the GC–MS

The evaluation of red pigment was performed according to the same methodology and conditions that were described by Girija et al., [21] with modification. In brief, the pigment was dissolved in hexane (1:20 *v/v*) and split injected Agilent 7890 gas chromatograph, the capillary column (30 m × 0.25 mm diameter × 0.25 µm). The operation conditions were 280 °C for injecting temperature, 45 °C for column oven temperature, and the helium flow was 1.40 mL/min. The overall holding time of 50 min. Mass spectra conditions were at electron impact (40 eV, 200 °C ion source temperature, interface temperature 240 °C.

### 2.9. Determination of Red-Pigment Content Using the FTIR

Fourier transforms infrared spectroscopy is used for the determination of the bonds starching vibration present in the extract component. The FTIR measuring was done by the same methodology described by Trivedi et al. [22]. In brief, the methanolic extract of pigment with optical density (1.5) was taken for FTIR analysis. A quantity of 10 µL of pigment was placed on a diamond window of the spectrophotometer under standard room temperature. A 32 scans with a resolution of 4 cm^−1^ was adapted. The FTIR spectrum was detected by the Perkin Elmer instrument in the range of 450–4000 cm^−1^ at a resolution of 4 cm^−1^.

### 2.10. Manufacturing of Meat Products

Two types of meat products were prepared: meatballs and burgers. The preparation of meatballs had been done according to the methodology described by Turhan et al. [23], where the burger was processed according to the method of Essa et al., [24]. The minced meat was fortified by red pigment at 3 mg/g of meat. This ratio was chosen according to the antifungal result of the minimal antifungal concentration (MFC) value. The MFC was recorded at 2.75 mg/g against *A. flavus* fungi, which are considered the most hazardous fungi for food products. Additionally, this application quantity (3 mg/g of meat) was recorded higher than the minimal inhibition concentration (MIC) value that was recorded against pathogenic bacterial strains in the antibacterial evaluation section of this study. The pigment implementation at this amount in minced meat also reflects an amelioration of meat red coloring, which could be optically observed. In this regard, the pigment application at this quantity could achieve two dimensions benefits, color enhancement, and safety properties in food model applications.

### 2.11. Color Measurements of Meat Products

The color of meatballs and burger samples were measured using a spectro-colorimeter (tristimulus color machine) with the CIE lab color scale (Hunter, Lab Scan XE, Reston VA.) Calibrated with a white standard tile of the Hunter Lab color standard (LXNO. 16379): X = 77.26, Y = 81.94, and Z = 88.14. The illuminant that was used in the measurement was the type A illuminant, the observer had the tristimulus color-mixture data recommended by the CIE for a large 10° field, while the area of viewing aperture was 25 mm. Measuring of samples were done in triplicates. Color difference (**ΔE**) was calculated from a, b, and L parameters, using the Hunter–Scotfield’s equation [25] as follows:**ΔE = (Δa^2^ + Δb^2^ + ΔL^2^)1/2**(1)
where **Δa** = **a** − **a_0_**, **Δb** = **b** − **b_0_**, and **ΔL** = **L** − **L_0_**, where the subscript “0” indicates the color of the control. The Hue angle (tg − 1 b/a) and saturation index (√a2 + b2) were calculated.

### 2.12. Texture Properties of Meat Products

Texture parameters (hardness, adhesiveness, springiness, cohesiveness, gumminess, and chewiness of meat products were measured objectively by using a texture analyzer TA-CT3 (AMETEK Brookfield, Delaware St., Chandler, AZ, USA), A cube sample (2 cm × 2 cm × 2 cm) was cut from a sample middle, placed centrally beneath the probe in the presence of the TA-BT-kit table. Cooked samples at room temperature were compressed axially in two consecutive cycles of 50% compression using an aluminum cylinder probe P/36 (36 mm diameter, Brookfield Engineering, Middleboro, MA, USA). Data collection and calculations were performed using the Texture Pro Software (Brookfield Engineering, CT3-50K, Middleboro, MA, USA). Force–time deformation curves were obtained with a 50 kg load cell applied at a cross-head speed of 2.0 mm.s^−1^. A trigger force of 5 g was applied. Analyses were performed in triplicate as described by Simone et al., 2013 [26].

### 2.13. Organoleptic Evaluation for Meat Samples

An organoleptic evaluation of samples was conducted for the steamed and fried products by 15 panelists from the staff members from the Department of Food Technology, National Research Centre, Egypt. The sensory evaluation as described by O’Mahony [27] for symmetry of shape (20), color (20), texture (20), aroma (20), and mouthfeel (20).

### 2.14. Determination of Pigment Impact on the Shelf Life

The extended time in the shelf life of the samples evaluated according to the methodology of Badr et al. [28] with a modification as follows: instead of doing infection for the samples, the increment in the total colony count was used as an indicator for decreasing of shelf life of meat products. The relation was expressed between the number of storage days and the logarithmic colony count. The lower logarithmic expressed the long life of the product in the same storage period.

### 2.15. Statistical Analysis

All experiments were performed in triplicate; data were represented as mean ± SD. The obtained data were subjected to the one-way analysis of variance (ANOVA procedure) using SPSS program ver. 25, IBM Co., Madison Ave, New York, United States.

## 3. Results

The present pigment of *P. purpurogenum* 2603 fungi was produced under the aseptic condition of nutrient sources, salinity, pH, and temperature that were previously investigated by Ahmed et al. [11]. Besides, the production stability was investigated and the production process reported better results compared to the other natural color sources. Whereas this red pigment reflects good characteristics and water solubility, it was recommended to implement in food products as a natural color source. In this case, more pigment characteristics were suggested to evaluate included the toxicity. After the evaluation, it had been used in meat products to ameliorate their color naturally. To the authors’ knowledge, it is the first application of the current natural red pigment in food products, which reflect the novelty in this research. Additionally, the authors have evaluated the safety properties and the stability of the pigment to indicate the suitability for food application.

### 3.1. Antibacterial and Antifungal Properties of the Red Pigment

The minimal antibacterial concentration (MIC) of the applied red pigment, which was extracted previously from the fungal media, was recorded at 320 µg/g against *E. coli* and *Salmonella* strains. This result represents good antibacterial properties in comparison to the standard antibacterial compounds. Moreover, the minimum antifungal concentration (MFC) was recorded at 2.75 mg/g against *A. flavus* fungi. These results showed a pigment efficacy against pathogenic microorganisms, which may participate in increasing safety by taking part in food products.

Furthermore, diffusion assays were used to ensure the antimicrobial of the red pigment. Using four solvent systems, the pigment was extracted and the extracts were evaluated against bacterial strains (G+ and G−) and toxigenic fungi. The results were expressed by higher efficacy, particularly for the non-polar extracts (Table 1). The inhibition zones against (G+) bacteria were recorded greater than that were recorded against (G−) strains. This could explain due to the high resistance of the (G−) that is commonly known. The inhibition zone using diffusion assay showed also an effective role of pigment extractions as an antifungal. The highest inhibition zone against fungal strains has been recorded against *Fusarium culmorum* where the lowest impact was recorded against the *Aspergillus flavus* ITEM 698 strain.

The microbial metabolites of various microorganisms reported antibacterial and antifungal properties [16,29]. It could be conjugated to active components that are secreted during the metabolism of the microorganisms [6]. These active components, either companion to pigment or one of its components, could describe the antifungal and antibacterial behavior of red pigment applied [30,31].

### 3.2. The Impact of Pigment on Mycotoxin Reduction

The red pigment impact at 1, 3, and 5 mg/mL broth media on mycotoxin production was determined. The reduction of aflatoxins and ochratoxin A was evaluated using the HPLC apparatus as described by Shehata et al. [16]. According to the results that were shown in Figure 1, aflatoxin B_1_ (AFB_1_) reduction ranged between 9 and 38%, for 1 mg/mL and 5 mg/mL, respectively. This ratio was elevated to reach 45% for 5 mg/mL pigment against AFB_2_. Where, ochratoxin reduction was recorded at 15%, 31%, and 54% for the pigment concentrations of 1, 3, and 5 mg/mL in media, respectively.

### 3.3. Toxicity Assay of Red Pigment

The toxicity assessment of the extracted pigment was varied due to the type of applied solvent. It was recorded at 8500 ppm for isopropanol and water, where it was 6400 ppm for petroleum ether and 5900 ppm for chloroform. This preliminary result represents the safety of pigment application. A further study of experimental animals might be demanded in future research.

In comparison to the previous studies on the red pigment extracted from fungi, the results reflect a high safety to apply in food products. As mentioned by Lagashetti et al. [25], the toxicity of red pigment from *P. purpurogenum* was shown as safe and was previously applied in some industrial processes included pharmaceuticals and cosmetics.

### 3.4. The Evaluation of Red Pigment by the GC–MS

The chromatogram of red-pigment analysis displays distinguished compounds, which could play bioactive roles by their application in food ingredients. According to the library analysis, these compounds were inclusive the derivatives of 2,3-butanediol, octanoic, and hydroxyl octanoic acid; glycerol; D-eyrthro-pentonic acid; ribitol; deoxyribolactone; 2-deoxy-pentofuranose; D-galactose; lyxose; hexadecanoic acid, cyclopenten-1-yl; 7 tetradecen-1-ol, 9-dodecyn-1-ol; 9, 12-octadecadienoic acid; oleic acid; linolenic acid; and alpha-linolenic acid (Figure 2). Lipid derivatives were presented in considerable amounts, represented by glycerol, oleic acid, linolenic acid, and alpha-linolenic acid. While some derivatives were possessed antibacterial and antifungal activity, like 2,3-butanediol and D-eyrthro-pentonic acid [27]. These results gave evidence for the antimicrobial properties recorded in the applied assays.

### 3.5. Determination of Red-Pigment Content Using the FTIR

The chromatogram of the FTIR has reported a high presence of carbohydrate fraction in the pigment extract. It was represented by the peak number 20. The focusing of the area of the specific carbohydrates curves was ranged from 1300 to 800 cm^−1^. The variety of changes of pigment extract is observed clearly in Figure 3. Referring to this area, it was represented the suggested amount of the oligosaccharides, which have a linkage between sugar molecules [32].

Numerous fractions are represented by this significant area, which is linked to the recorded antimicrobial activity of the pigment. The expanded oscillations 1030–1075 cm^−1^ pointed to the fractions resulted from Fructan [33], this is more clearly observed. While, expanded oscillations 1240–1280 cm^−1^ (peak no.18) are linked to the aromatic fractions of hemicellulose and lignin [34,35], The expanded oscillation of 1600 and 1700 cm^−1^ in the red-pigment were connected to the fiber fractions, which were resulted from water-binding by the fiber hydrophilic group [34,35,36]. The expanded oscillation of 2900 cm^−1^ is joined to an alkyl group of (C–H) in red-pigment, while the oscillations of 3200–3800 cm^−1^ intensity of the bands were referred to the (-OH groups), which distinguished between polysaccharides and oligosaccharides [36,37].

### 3.6. Color Measurements of Meat Products

The manufactures of processed meat have a serious concern to attract the consumer’s interest, particularly for the final color of the product. The stability and shining of red color are requested for meat and meat products to realize the consumer expectations. Otherwise, the texture characteristics may give food products the prospective properties that meet consumer needs and requirements to tolerate transportation and handling conditions. The application of red pigment in meatballs and burgers (either steamed or fried cooked) reflected significant changes in the final product (Table 2). The changes were joined to the pigment insertion in the recipe of meat products. This effect was clear in hardness properties where treated samples were recorded softer. The chewiness of samples showed by increments in the treated samples, these increases were also reported in the previous works [38].

Moreover, the results in Table 2 showed an improvement of the color attributes of meat products by application of the red pigment in the recipe. The (**L***) values in colored products in both steamed and fried ones were recorded less than the non-treated samples by pigment. For the values of (**a***); it was recorded with high values compared to the non-treated samples of meat products. These results indicate the improvement, which has happened in the treated meat product samples. Additionally, it represents stability for the red pigment that is applied to the product during the cooking process.

The changes, which were recorded between fried and steamed products could be joined with the cooking process. The hardness changes were not significant in the final products except for fried colored burgers. It was observed clearly that the springiness of colored meat products was ameliorated compared to the non-treated ones. For the chewiness, it was non-significant except for fried burgers. Generally, it could be concluded that the treatment of minced meat products by the red microbial pigment enhanced their texture characteristics besides the color amelioration.

The earlier studies pointed to the fungal red pigments application to improve color, such as its implementation in the textile dyes. This pigment was reported stable and could produce in a large amount due to the growth conditions in fungal media [11].

### 3.7. Organoleptic Evaluation of Meat Samples

The organoleptic characteristics were evaluated by the expertise of food science from five foundations related to food research, food-products development, and food manufacturing. The evaluation experiment had been done according to the ethics and procedures described by Kilcast [39]. The results in Table 3 showed significant differences between the treated and non-treated meat samples. The differences were shown more clearly in the color, texture, and taste of fried products.

It was recorded that amelioration in color and taste of treated samples by red pigment, despite fine reductions shown in the texture properties of the final product. The overall opinion of the panelist impression of this treatment was accepted by 85% of the panelist-board. They explained the acceptable as the color and taste enhancement is more preferable for the customer, and texture changes were not far from the control.

### 3.8. Determination of Pigment Impact on Meat-Products Shelf Life

The shelf life impact, according to the presence of fungal-red pigment in the minced meat product recipes, was investigated. The extension of shelf life is represented at the lowest in the logarithmic (log) of total colony count (CFU) of microorganisms on the product during the cold storage period. Forasmuch as the two types of minced meat products (meatballs and burgers) are kept in the same storage conditions (25 °C), the authors suggested making five replicates for the burger products, where changes are represented as the mean of the CFU value, it also was considered evidence for changes that could be happened for meatballs. As shown in Figure 4, the positive effect of pigment in the minced meat products was recorded. The log CFU has shown lower in color minced meat either it was cooked by steaming or by frying. The more the storage period, the difference of log CFU recorded increased between treated and untreated samples.

## 4. Discussion

Minced meat and its related products have the interest of meat consumers, this could relate to the additives that give wonderful taste and aroma. Moreover, color is also one of the meat organoleptic attributes that consumers search for. Minced meat products include burgers and meatballs, in which spices are the main ingredients. The mycotoxin hazard in the products that are based on the minced-meat is joined with two sources; the tissue of muscles itself [9] or the spices [40], which was added in recipes. Ozonation [41] of spices could be solved for mycotoxin contamination of species before it was applied in food products. While, various extracts and metabolites achieved mycotoxin reduction in food products [7,15,16,42], mainly was depending on the antioxidant potency, metabolites, and phenolic acid content.

Mycotoxins are considered a real issue in various types of processed meat. Several studies referred that cured and processed meats can be contaminated with toxigenic fungal strains, essentially if they were processed in a warm climate [43,44,45,46]. The processing conditions during meat aging probably support the aflatoxin synthesis [47,48]. Indeed, it seems that there is no association between the fungal presence and aflatoxin contamination of meat. The frequent contamination of spices and additives utilized in meat processing might also represent a source of mycotoxin contamination [49]. The usage of contaminated spices as an ingredient in processed meat resulted in aflatoxins contamination of final processed meat. Recently, some approaches were applied successfully in mycotoxin degradation using bioactive components [15,42].

The color has a special interest of consumers in meat products, they always search for a product with enrichment color, which gives an impression for more muscles than fats. The darker the red product, the more preferred choice for buying. Natural pigments are more preferable recently for applying to food products. Animal and plant sources of pigments suffering from instability and insufficient, however, microbial pigment reported solving these disadvantages [6,30]. Microbial metabolites, besides providing natural pigments, these components have various therapeutic functions such as antioxidant, anticancer, and antitumor agents. For natural pigments, although it was produced as secondary metabolites, it controlled by special regulation inside the microbial system. These metabolites could achieve protection, particularly against food hazards and mycotoxins [50,51]. Hence, these compounds seem to preserve the products in which it was fortified with, besides that essential purpose it was applied for. Otherwise, these pigments are fermentative, which are affected by growth conditions. The usage of these pigments is approved for humankind [30].

Fungi could produce secondary metabolites with beneficial properties. These metabolites include carotenoids, antioxidants, and anthocyanins where all of which could also serve as natural pigments [52]. These metabolites were reported by antimicrobial potency [8,42]. By studying the antimicrobial characteristics of *P*. *purpurogenum* pigment, it was recorded possessing antibacterial and antifungal potency against the pathogenic bacteria and toxigenic fungal strains in the present study. The pigment extract practices mycotoxin reduction against aflatoxins and ochratoxin A in broth media. The maximum reduction was recorded between 38 and 45% for aflatoxins and 54% for ochratoxin A. This pigment also showed save in toxicity assay up to 8 g/kg for isopropyl extract.

Concerning *P*. *purpurogenum* pigment applications, it was inserted here in meatball and burger manufacturing, this was to evaluate its sensorial impact on real products. Color attributes exhibited amelioration in minced meat products cooked by two methods (steamed and fried). Texture parameters were recorded by a few variations like increases of the chewiness in treated samples. The panelist board of organoleptic-evaluation recommended pigment application for its impact on color and taste, although fine texture changes were taking place. Moreover, the insertion of red pigment in meat products showed an increment of shelf life that was expressed as decreases of CFU on treating samples during the storage period, which was in agreement with the previous investigations [53,54].

## 5. Conclusions

The modern trend of food enhancement is including the application of natural colorants, which could achieve extra benefits although of their relative instability. In this regard, the present investigation focused on the application of natural pigments to achieve the consumer expectation and more safety for the meat products. A red pigment was isolated from *P. purpurogenum* for application in two types of meat products. The pigment was analyzed using GC–MS and FTIR apparatus to reflect its enrichment of bioactive molecules. By evaluating the pigment extracts, it was reflected an antibacterial, antifungal, and antimycotoxigenic impacts. The safety impacts of the pigment for targeted applications were evaluated as its capacity to reduce mycotoxin secretion by fungi in liquid media. Toxin-secretion reduction using pigment extract in fungal media was recorded up to 38%, 45%, and 54% for AFB_1_, AFB_2_, and OCA, respectively. Safety characteristics of pigment recorded free of citrinin and cytotoxic behavior. Fortification of meatballs and burgers using this pigment represents an amelioration in meat sensoria, organoleptic, and texture, also enhanced their shelf life throughout the evaluation of total count increment during the storage period. It can conclude, application of *P. purpurogenum* pigment in meat fortification reflect an amelioration in sensorial, shelf life, and antimicrobial properties, side to the reduction recorded for mycotoxin levels in liquid fungal media.

## Figures and Tables

**Figure 1 microorganisms-09-00989-f001:**
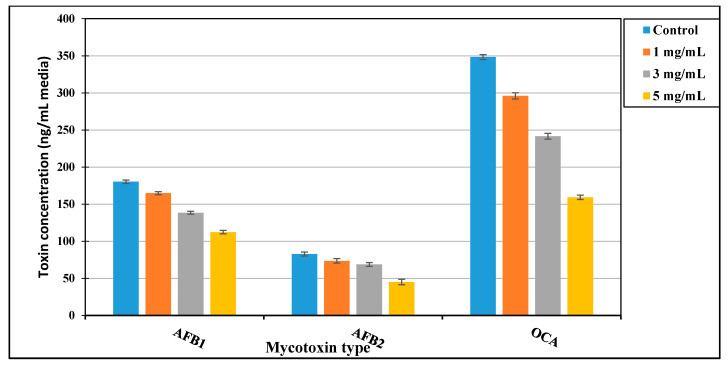
The pigment impact on aflatoxins and ochratoxin A reduction. AFB_1_: aflatoxin B_1_; AFB_2_: aflatoxin B_2_; OCA: ochratoxin A. (1 mg/mL; 3 mg/mL; 5 mg/mL) are pigment concentration in media.

**Figure 2 microorganisms-09-00989-f002:**
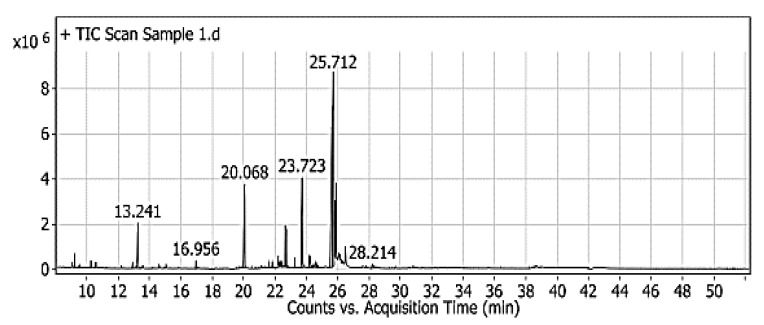
The GC–MS chromatogram of red-pigment analysis that was extracted from *P. purpurogenum.*

**Figure 3 microorganisms-09-00989-f003:**
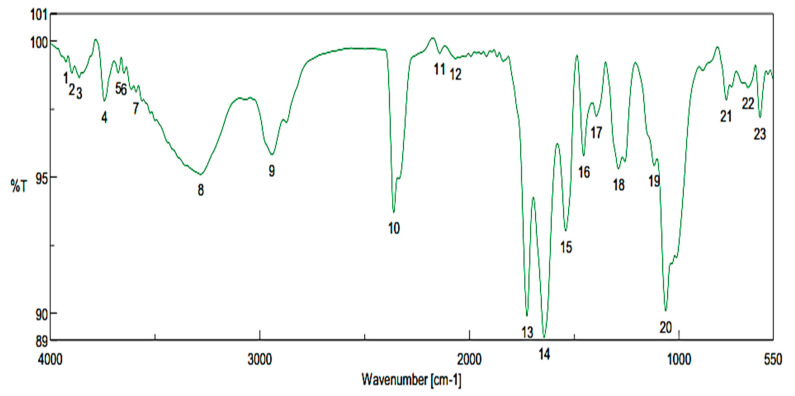
The FTIR-chromatogram of red-pigment analysis extracted from *P. purpurgenum.*

**Figure 4 microorganisms-09-00989-f004:**
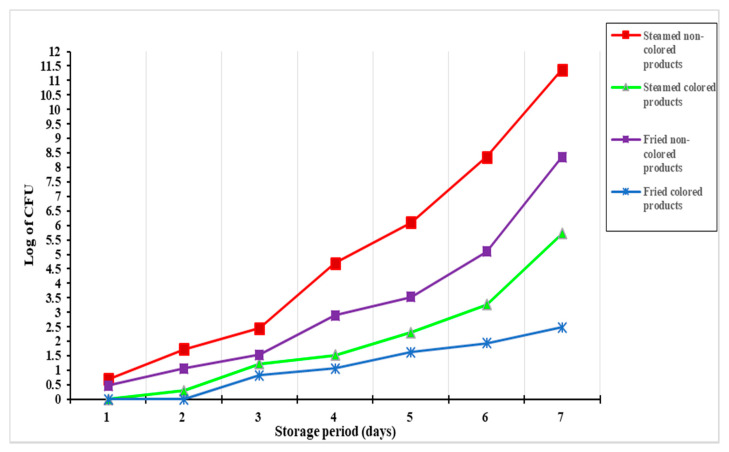
Enhancement of the shelf life of meat products using fungi pigment.

**Table 1 microorganisms-09-00989-t001:** The inhibition zones of pigment extracts as antimicrobial agents.

Microorganisms	Inhibition Zone (mm)			
	Ethyl Acetate Extract	Chloroform Extract	Iso-Propyl Extract	Methanol Extract
**Bacterial strains**				
*Bacillus cereus* ATCC 4342	7.8 ± 0.7 ^a^	4.1 ± 0.5 ^d^	8.2 ± 1.0 ^b^	6.9 ± 1.1 ^c^
*Staphylococcus aureus NCTC 10788*	6.4 ± 0.9 ^a^	3.7 ± 0.4 ^c^	6.7 ± 0.7 ^a^	7.1 ± 0.6 ^b^
*Salmonella typhi ATCC 14028*	0.5 ± 0.01 ^a^	0.4 ± 0.02 ^b^	0.3 ± 0.01 ^c^	0.3 ± 0.08 ^c^
*Escherichia coli ATCC 11229*	0.4 ± 0.03 ^a^	0.2 ± 0.01 ^c^	0.3 ± 0.01 ^b^	0.3 ± 0.01 ^b^
**Fungal strains**				
*Aspergillus flavus* ITEM 698	1.3 ± 0.4 ^a^	1.5 ± 0.6 ^b^	1.8± 0.9 ^c^	1.6 ± 0.7 ^b^
*Aspergillus ochraceus* ITEM 5010	1.8 ± 0.3 ^a^	1.8 ± 0.7 ^a^	1.7 ± 0.5 ^b^	1.9 ± 0.4 ^a^
*Aspergillus niger* ITEM 3856	2.2 ± 1.0 ^a^	2.4 ± 0.9 ^b^	2.3 ± 0.6 ^c^	2.3 ± 0.8 ^c^
*Fusarium culmorum* KF846	3.2 ± 1.2 ^a^	3.0 ± 0.7 ^b^	3.2 ± 1.0 ^a^	3.5 ± 0.5 ^c^

The data expressed as means ± SD (where *n* = 3 at *p* = 0.05). The letter that different in each row means significant differences.

**Table 2 microorganisms-09-00989-t002:** Texture properties and color improvement of meat products processed by the microbial red pigment.

	Meatballs				Burgers			
	Non-Colored Steamed	Colored Steamed	Non-Colored Fried	Colored Fried	Non-Colored Steamed	Colored Steamed	Non-Colored Fried	Colored Fried
**Hardness (N)**	15.38 ± 1.05 ^a^	14.95 ± 1.48 ^a^	17.3 ± 1.71 ^b^	15.05 ± 1.37 ^a^	14.77 ± 1.41 ^a^	13.4 ± 1.36 ^b^	15.34 ± 1.22 ^a^	11.16 ± 1.27 ^d^
**Springiness (Mm)**	0.68 ± 0.13 ^a^	0.82 ± 0.14 ^b^	0.75 ± 0.27 ^c^	0.9 ± 0.21 ^d^	1.13 ± 0.04 ^a^	1.24 ± 0.03 ^b^	1.9 ± 0.88 ^c^	2.01 ± 0.93 ^d^
**Cohesiveness**	0.63 ± 0.11 ^a^	0.85 ± 0.24 ^b^	0.7 ± 0.06 ^a^	0.81 ± 0.18 ^b^	0.75 ± 0.37 ^a^	0.62 ± 0.18 ^a^	0.63 ± 0.17 ^a^	0.8 ± 0.16 ^b^
**Chewiness (mJ)**	7.12 ± 0.97 ^a^	7.3 ± 1.67 ^a^	7.2 ± 0.54 ^a^	7.91 ± 0.93 ^a^	7.9 ± 1.14 ^a^	8.45 ± 1.26 ^a^	9.5 ± 1.02 ^b^	9.7 ± 1.16 ^b^
**Gumminess (N)**	6.68 ± 1.09 ^a^	6.13 ± 1.51 ^a^	8.43 ± 0.87 ^b^	8.26 ± 1.08 ^b^	29.6 ± 1.64 ^a^	25.1 ± 1.81 ^b^	27.3 ± 1.48 ^c^	22.9 ± 1.54 ^d^
**L ***	45.12 ± 1.05 ^a^	36.33 ± 1.36 ^b^	45.15 ± 1.22 ^a^	31.36 ± 1.41 ^c^	40.12 ± 1.79 ^a^	34.16 ± 1.59 ^b^	36.63 ± 1.81 ^c^	30.88 ± 1.54 ^d^
**a ***	14.78 ± 1.21 ^a^	25.91 ± 1.15 ^b^	12.57 ± 1.26 ^a^	27.87 ± 1.84 ^c^	14.37 ± 1.09 ^a^	24.88 ± 1.37 ^b^	13.24 ± 1.22 ^a^	25.41 ± 1.37 ^b^
**b ***	12.27 ± 1.74 ^a^	27.24 ± 1.63 ^c^	14.82 ± 1.27 ^a^	19.42 ± 1.66 ^b^	17.31 ± 1.54 ^a^	27.43 ± 1.28 ^d^	13.27 ± 1.07 ^b^	20.55 ± 1.93 ^c^

The data expressed as means ± SD (where *n* = 3 at *p* = 0.05). (L *) represents lightness; (a *) represents redness; where (b *) represents yellowness. For meatballs; the letter that different in each row means significant differences. For burgers; the letter that different in each row means significant differences.

**Table 3 microorganisms-09-00989-t003:** Organoleptic evaluation of meat products, processed by the microbial red pigment.

Samples	Color (20)	Shape (15)	Texture (15)	Aroma (15)	Taste (15)	Mouthfeel (15)
**Meatballs**						
**Steamed non-colored products**	17.4 ^a^	13.3 ^a^	13.1 ^a^	13.8 ^a^	13.2 ^a^	14.4 ^a^
**Steamed colored products**	19.6 ^b^	13.7 ^a^	13.3 ^b^	13.9 ^a^	13.2 ^a^	14.5 ^a^
**Fried non-colored products**	16.7 ^a^	13.4 ^a^	11.1 ^c^	14.2 ^b^	13.8 ^b^	14.2 ^a^
**Fried colored products**	19.5 ^b^	14.1 ^b^	11.5 ^d^	14.3 ^b^	14.4 ^c^	14.6 ^a^
**LSD at 0.05**	1.185	0.701	0.241	0.522	0.628	0.819
**Burgers**						
**Steamed non-colored products**	18.2 ^a^	12.8 ^a^	13.8 ^a^	13.7 ^a^	13.5 ^a^	14.2 ^a^
**Steamed colored products**	19.8 ^b^	14.3 ^c^	14.5 ^b^	13.8 ^a^	13.6 ^a^	14.6 ^b^
**Fried non-colored products**	18.1 ^a^	13.5 ^b^	13.9 ^a^	14.2 ^b^	14.1 ^b^	14.3 ^a^
**Fried colored products**	19.7 ^b^	14.5 ^c^	14.7 ^c^	14.3 ^b^	14.2 ^b^	14.5 ^b^
**LSD at 0.05**	1.573	0.491	0.197	0.218	0.257	0.305

The data expressed as means (where *n* = 3 at *p* = 0.05). The letter that is different in each column, for the individual product, means significant differences.

## Data Availability

All data are available in the present article and no other data in this study are available.

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
