# Peer review of "New Antifungal Microbial Pigment Applied to Improve Safety and Quality of Processed Meat-Products"

_microorganisms, 2021, doi:10.3390/microorganisms9050989_

Round 1

Reviewer 1 Report

The article is prepared correctly. Minor inaccuracies and editorial errors do not detract from the meaning of the work. However, I have the impression that this manuscript should be presented in a journal about food.

No statistically significant differences were identified in tables 1 and 2, despite the fact that the table description states that ... "The data expressed as means ± SD (where n = 3 at p = 0.05)" ... The authors are asked to complete and indicate significant differences.

Please review all references and correct them as required by the journal. Moreover, 19 items out of 50 are cited from 10 years ago. The authors are asked to refresh the cited reports.

Author Response

Reviewer 1:

The conclusions can be improved to support by the results.

Thanks, this point was covered in the revised copy.

No statistically significant differences were identified in tables 1 and 2, despite the fact that the table description states 

Thanks, this point was covered in the revised copy.

Please review all references and correct them as required by the journal.

Thanks, this point was covered in the revised copy.

Reviewer 2 Report

Review ID: microorganisms-1191217

The reviewed Ms ID: microorganisms-1191217 in a first glance is an interesting and the manuscript is well-organized. However, a precise look at the text reveal that the paper is not well written, methodologically flawed and showing many misunderstandings, unintentional, I hope. In my opinion, the subject of this manuscript is in general suitable for the Journal; however, in present form I cannot recommend this article for publication due to scientific and editorial reservations.

Main comments

1. Whether indeed the pigment used in the study can be considered 'new' and thus the study can be described as novel. This requires justification by the authors.

2. The authors should clearly justify why they assumed a red pigment concentration of 3 mg/g meat in their study. The information provided (L153-154) is insufficient. 

3. I have serious reservations about the methodology adopted for texture assessment. Why was the method used for starch-based food products (firmness of bread) and not dedicated to meat and meat products? Furthermore, it is not stated which test and probe were used.

4. The statistical analysis is insufficiently presented and needs to be described in detail. There are no models described for the analysis of the data. Information on the number of samples and replicates was also not provided.

5. It is incomprehensible (and there is no explanation) why some results (colour and texture) are given separately for meatballs and burgers, while the results of organoleptic evaluation and shelf life are given for the enigmatic ‘meat products’. Was it a third product or were meatballs and burgers included together. So what statistical methods were used?  A proper statistical method entitles the authors to justified conclusions.

Other remarks

L83-86: Provide a complete methodology of pigment extraction

L86: provide the name and manufacturer of an instrument

L95: the numbering of references should be updated taking into account Abdel-Razek et al. 2017 as item [12], which was omitted in References 

L101-107: Provide a complete methodology with appropriate literature method

L129, 134, 138, 160: I did not find these authors in References

L146-148: Provide a complete methodology with appropriate literature method, how many scans for each sample, temperature of measurement etc.

L156-160: Please, provide information about an illuminant, observer, and area of viewing aperture

L180185: what conditions were adopted and what temperature was used (4° or 25°C)

Table 1

  • Were significant differences found between the averages? (p=0.05)
  • Please check value 7.3 for Salmonella and Methanol extract

Table 2

  • Were significant differences found between the averages? (p=0.05)

The list of references should be updated.

Other suggestions 

L54: ‘several’ replace with ‘series’, in order to avoid repetition

L57: add ‘of food’ at the end of sentence.

L63: should be ‘the’

L73: 250 mL

L148: provide manufacturer of an instrument

L164-165: correct the explanations to the formula

L282: expectations rather than wishes

Author Response

Reviewer 2:

Main comments

  1. Whether indeed the pigment used in the study can be considered 'new' and thus the study can be described as novel. This requires justification by the authors.

A paragraph was added to describe that as:

The present pigment of P. purpurogenum 2603 fungi was produced under the aseptic condi-tion of nutrient sources, salinity, pH, and temperature that were previously investigated by Ahmed et al.,[10].  Besides, the production stability was investigated and the production pro-cess reported better results compared to the other natural color sources.  Whereas this red pigment reflects good characteristics and water solubility, it was recommended to implement in food products as a natural color source. In this case, more pigment characteristics were sug-gested to evaluate included the toxicity. After the evaluation, it had been used in meat products to ameliorate their color naturally. To the authors’ knowledge, it is the first application of the current natural red pigment in food products, which reflect the novelty in this research. Also, the authors have evaluated the safety properties and the stability of the pigment to indicate the suitability for food application.

  1. The authors should clearly justify why they assumed a red pigment concentration of 3 mg/g meat in their study. The information provided (L153-154) is insufficient.

This point was corrected in the paragraph as:

The minced meat was fortified by red pigment at 3 mg/g of meat. This ratio was chosen accord-ing to the antifungal result of the minimal antifungal concentration (MFC) value. The MFC was recorded at 2.75 mg/g against A. flavus fungi, which are considered the most hazardous fungi for food products. Also, this application quantity (3 mg/g of meat) was recorded higher than the minimal inhibition concentration (MIC) value that was recorded against pathogenic bacterial strains in the antibacterial evaluation section of this study. The pigment implementation at this amount in minced meat also reflects an amelioration of meat red coloring, which could be optically observed. In this regard, the pigment application at this quantity could achieve two dimensions benefits, color enhancement and safety properties in food model applications.       

  1. I have serious reservations about the methodology adopted for texture assessment. Why was the method used for starch-based food products (firmness of bread) and not dedicated to meat and meat products? Furthermore, it is not stated which test and probe were used.

This part of methodology was written by a typographical error, it was rewritten as the methodology done by right method as:

Texture parameters (hardness, adhesiveness, springiness, cohesiveness, gumminess, and chewiness of meat products were measured objectively by using a texture analyzer TA-CT3 (Brookfield, USA), A cube sample (2 x 2x 2cm) was cut from a sample middle, placed centrally beneath the probe in the presence of the TA-BT-kit table. cooked samples at room temperature were compressed axially in two consecutive cycles of 50% compression using an aluminum cylinder probe P/36 (36 mm diameter, Brookfield Engineering, Middleboro, MA, USA). Data collection and calculations were performed using the Texture Pro Software (Brookfield Engineering, CT3-50K, Middleboro, MA, USA). Force-time deformation curves were obtained with a 50 kg load cell applied at a cross-head speed of 2.0 mm.s-1. A trigger force of 5 g was applied. Analyses were performed in triplicate as described by Simone et al., 2013 [23],

  1. The statistical analysis is insufficiently presented and needs to be described in detail. There are no models described for the analysis of the data. Information on the number of samples and replicates was also not provided.

This part was corrected in the revised copy of the manuscript.

The significance of differences was provide in complete model for Table 1 and 2,

The analysis model was done using analysis of variances (ANOVA one way)

All experiments were performed in triplicate; data were represented as mean ± SD. The obtained data were subjected to the one-way analysis of variance, (ANOVA procedure) Using SPSS program ver. 25, IBM Co., USA

  1. It is incomprehensible (and there is no explanation) why some results (colour and texture) are given separately for meatballs and burgers, while the results of organoleptic evaluation and shelf life are given for the enigmatic ‘meat products’. Was it a third product or were meatballs and burgers included together. So what statistical methods were used? A proper statistical method entitles the authors to justified conclusions.

 Thanks for this refer point, it is really important

The authors were correct this point in the revised version of manuscript

And the missed Data were added in correct way with the statistics.

For the shelf life, author suggested that, whereas the two products may store freshly in the same condition, they could be evaluated as one product, and this could reflect the changes may happen for the other. This point was written as follow in revised manuscript:

            The shelf life impact, according to the presence of fungal-red pigment in the minced meat product recipes was investigated. The extension of shelf life is represented at the lowest in the logarithmic (log) of total colony count (CFU) of microorganisms on the product during the cold storage period. Forasmuch the two types of minced meat products (meatballs and burgers) are kept in the same storage conditions (23 ºC), the authors suggested making five replicates for the burgers products, where changes are represented as the mean of CFU-value, it also was considered evidence for changes that could be happened for meatballs. As shown in Fig. 4, the positive effect of pigment in the minced meat products was recorded. The log CFU has shown lower in color minced meat either it was cooked by steaming or by frying. The more the storage period, the difference of log CFU recorded increased between treated and untreated samples.

Other remarks

L83-86: Provide a complete methodology of pigment extraction

Pigments Extraction

The culture was shaken at 200 rpm (28°C / 7 days). The production medium was pre-pared using the following nutrients (g/l distilled water): Mannitol (20), NaNO3 (3.0), KCl (0.5), MgSO4.7H2O (0.5), KH2PO4 (1.0), FeSO4.7H2O (0.1). Cultures from Dox production media were filtered through filter paper (No. 1; Whatman), the biomass was dried at 50ºC for 48 h and weighed. The red pigment was extracted with Chloroform, Methanol, Isopropanol, and Ethyl acetate. The extract was concentrated till dryness and dissolved in 10 ml distilled water for application in characteristics evaluations [10]. The red pigment was further extracted by isopropanol for food applications experiments, evaporated, concentrated till dryness using the rotary evaporator (Heidolph, Hei-VAP Platinum 5; ProfiLab24 GmbH; Landsberger; Germany. The absorbance was read at 630 using a UV-visible spectrophotometer ( Spectro UV-2505; La-bomed Co., Int., La Cienega Blvd, Los Angeles; USA).

L86: provide the name and manufacturer of an instrument

rotary evaporator (Heidolph, Hei-VAP Platinum 5; ProfiLab24 GmbH; Landsberger; Germany. The absorbance was read at 630 using a UV-visible spectrophotometer ( Spectro UV-2505; La-bomed Co., Int., La Cienega Blvd, Los Angeles; USA).

L95: the numbering of references should be updated taking into account Abdel-Razek et al. 2017 as item [12], which was omitted in References

Thanks, this point was covered in the revised copy.

L101-107: Provide a complete methodology with appropriate literature method

    In broth media of fungal growth, anti-mycotoxin properties of the red pigment were evaluated according to the method described by Shehata et al. [15], using the pigment extract (1, 3, and 5mg/mL) against fungal growth of Aspergillus flavus ITEM 698 (AFs producer), and A. ochraceus ITEM 5010 (OCA producer). Briefly; in conical flasks contains autoclaved 150 mL of yeast extract sucrose media (YES) for fungal growth of applied strain. Each conical flask was inoculated by the fungal spores that was re-suspended in tween/water solution at a concentration of 10 5 spores/mL. Nine flasks were prepared for the three applied concentration of pigment extracts (1, 3, and 5mg/mL; three replicates per each) in comparison to the control flask, which only inoculated by fungal spores. This design was applied for the two fungal types under the investigation. The reduction of mycelial growth weight will reflect the antifungal activates, while the toxin producing reduction reflect the safety impact of the applied concentration. The AFs and OCA were recovered from the growth media, cleaned up using Aflatest and Ochratest columns, respectively. The reduction of mycotoxins joined with the pigment was determined according to the conditions, columns, and apparatus described in Shehata et al. [15].

L129, 134, 138, 160: I did not find these authors in References

Thanks, this point was covered in the revised copy.

L146-148: Provide a complete methodology with appropriate literature method, how many scans for each sample, temperature of measurement etc.

Fourier transforms infrared spectroscopy is used for the determination of the bonds starching vibration present in the extract component. The FTIR measuring was done by the same methodology described by Trivedi et al., [21]. In brief, the methanolic extract of pigment with optical density (1.5) was taken for FTIR analysis. A quantity of 10 µl of pigment was placed on a diamond window of the spectrophotometer under standard room temperature. A 32 scans with a resolution of 4 cm-1 was adapted. The FTIR spectrum was detected by the Perkin Elmer instrument in the range of 450 to 4000 cm-1 at a resolution of 4 cm-1.

L156-160: Please, provide information about an illuminant, observer, and area of viewing aperture

The color of meatballs and burger samples were measured using a Spectro-Colorimeter (Tristimulus color machine) with the CIE lab color scale (Hunter, Lab Scan XE, Reston VA.) Calibrated with a white standard tile of Hunter Lab color standard (LXNO. 16379): X= 77.26, Y= 81.94, and Z= 88.14. The illuminant that used in the measurement was type A illuminant, the observer have the Tristimulus color-mixture data recommended by the CIE for a large 10° field, while the area of viewing aperture was 25 mm. Measuring of samples were done in a triplicates. Color difference (ΔE) was calculated from a, b and L parameters, using Hunter-Scotfield’s equation.

L180185: what conditions were adopted and what temperature was used (4° or 25°C)

Thanks, this point was covered in the revised copy, and it was done at 25 °C

Table 1

Were significant differences found between the averages? (p=0.05).

Thanks, this point was covered in the revised copy.

Please check value 7.3 for Salmonella and Methanol extract.

Thanks, it was just a typographical mistake and it was corrected.

Table 2

Were significant differences found between the averages? (p=0.05).

Thanks, this point was covered in the revised copy.

The list of references should be updated.

Thanks, this point was covered in the revised copy.

Other suggestions

L54: ‘several’ replace with ‘series’, in order to avoid repetition

Thanks, this point was covered in the revised copy.

L57: add ‘of food’ at the end of sentence.

Thanks, this point was covered in the revised copy.

L63: should be ‘the’

Thanks, this point was covered in the revised copy.

L73: 250 mL

Thanks, this point was covered in the revised copy.

L148: provide manufacturer of an instrument

Thanks, this point was covered in the revised copy.

L164-165: correct the explanations to the formula

Thanks, this point was covered in the revised copy.

L282: expectations rather than wishes.

Thanks, this point was covered in the revised copy.

Round 2

Reviewer 2 Report

The corrected article was improved in accordance with a number of proposed suggestions.  Moreover, authors tried to provide completely explanations and exact answers for all requests.  I really appreciate their efforts and contribution to amended version of manuscript.

In my opinion the manuscript requires only a few editorial corrections before publication.

L81, L83, L93: should be .../mL; g/L 

Section 2.1. and 2.2.: numbers in names of chemical compounds should be in lower superscripts 

L116: '2017' is redundant  

section 2.11.: please add '*' with a, b, and L indices if values are given in CIE L*a*b* system

Fig. 1 and Fig. 4 should be more readable (they are just blurry).

I am sure that readers would be more satisfied with a more detailed description of the statistical analysis. Note to be considered by the authors.